# Simultaneous Quantification and Speciation of Trace Metals in Paired Serum and CSF Samples by Size Exclusion Chromatography–Inductively Coupled Plasma–Dynamic Reaction Cell–Mass Spectrometry (SEC-DRC-ICP-MS)

**DOI:** 10.3390/ijms22168892

**Published:** 2021-08-18

**Authors:** Bernhard Michalke, Achim Berthele, Vivek Venkataramani

**Affiliations:** 1Research Unit Analytical BioGeoChemistry, Helmholtz Center Munich—German Research Center for Environmental Health, 85764 Neuherberg, Germany; 2Department of Neurology, School of Medicine, Technical University of Munich (TUM), Klinikum rechts der Isar, 81675 Munich, Germany; achim.berthele@tum.de; 3Department of Medicine II, Hematology/Oncology, University Hospital Frankfurt, 60590 Frankfurt am Main, Germany; vivek.venkataramani@kgu.de; 4Institute of Pathology, University Medical Center Göttingen (UMG), 37075 Göttingen, Germany

**Keywords:** iron, manganese, element speciation, paired samples, cerebrospinal fluid, serum, SEC-ICP-DRC-MS

## Abstract

Background: Transition metals play a crucial role in brain metabolism: since they exist in different oxidation states they are involved in ROS generation, but they are also co-factors of enzymes in cellular energy metabolism or oxidative defense. Methods: Paired serum and cerebrospinal fluid (CSF) samples were analyzed for iron, zinc, copper and manganese as well as for speciation using SEC-ICP-DRC-MS. Brain extracts from Mn-exposed rats were additionally analyzed with SEC-ICP-DRC-MS. Results: The concentration patterns of transition metal size fractions were correlated between serum and CSF: Total element concentrations were significantly lower in CSF. Fe-ferritin was decreased in CSF whereas a LMW Fe fraction was relatively increased. The 400–600 kDa Zn fraction and the Cu-ceruloplasmin fraction were decreased in CSF, by contrast the 40–80 kDa fraction, containing Cu- and Zn-albumin, relatively increased. For manganese, the α-2-macroglobulin fraction showed significantly lower concentration in CSF, whereas the citrate Mn fraction was enriched. Results from the rat brain extracts supported the findings from human paired serum and CSF samples. Conclusions: Transition metals are strictly controlled at neural barriers (NB) of neurologic healthy patients. High molecular weight species are down-concentrated along NB, however, the Mn-citrate fraction seems to be less controlled, which may be problematic under environmental load.

## 1. Introduction

The first-row transition metals iron (Fe), manganese (Mn), zinc (Zn) and copper (Cu) can exist in multiple oxidation states and participate in electron transfer reactions that are fundamental to sustain life for all organisms that undertake oxidative metabolism. Approximately one-third of the human proteome is bound to metals [1,2]. Unlike post-translational protein modifications, such as glycosylation and phosphorylation that do not always have a one-to-one relationship with protein function [3], the presence of a metal co-factor can be directly linked with enzymatic function [4]. Indeed, trace elements functionally serve as suitable catalysts for crucial biological redox processes (e.g., respiration, metabolism and signaling processes), and structural components of essential bioactive molecules (e.g., iron-containing hemoglobin and zinc finger transcription factors [5,6]).

Especially the central nervous system (CNS) is sensitive to disturbances in trace element concentrations due to their extensive utilization in order to satisfy the high neuronal energy demand. Given the relatively narrow range between deficiency and toxicity, trace elements need to be tightly regulated to ensure optimal neuronal development and maintenance of healthy brain functions [7]. Regulation of trace element fluxes from the vascular system to the tightly sealed CNS compartment is accomplished by two neural barriers (NB): the blood–brain barrier (BBB) and the blood–cerebrospinal fluid barrier (BCB). While the BBB is localized at the level of endothelial cells within the brain vasculature, resulting in a physical barrier to material exchange between blood and brain tissues and fluid, the BCB is composed of ependymal cells that separate the blood from the ventricular cerebrospinal fluid (CSF) [8]. In this regard, NB shield the CNS from potentially harmful metals in the blood, but at the same time also supply brain tissue with essential trace elements via a series of active or receptor-mediated transport systems (e.g., receptor-mediated endocytosis of Fe-bound transferrin) to maintain their optimal concentrations [9,10].

Importantly, small amounts of free and “labile” redox-active elements, such as chemically reduced Fe and Cu, are able to catalyze reactive oxygen species (ROS) via Fenton chemistry, generating hydrogen peroxides, hydroxyl radicals, and superoxide anions. At physiological relevant scales, ROS act as pleiotropic signaling molecules (“ROS signaling” [11]). However, excess amounts of Fe and Cu can result in elevated and uncontrolled ROS formation causing macromolecular damage (e.g., proteins, lipids and nucleotides), denoted under the umbrella term “oxidative stress (OS)”. Disruption of metal homeostasis driving OS is strongly linked to cell death (e.g., apoptotic and non-apoptotic death [12,13]), ischemia-reperfusion damage (e.g., ischemic, hemorrhagic and traumatic brain trauma) and multiple neurodegenerative conditions including Alzheimer’s disease (AD), Parkinson’s disease (PD) and Huntington’s disease (HD) (reviewed in [14,15]). Hence, maintenance of brain redox homeostasis relies on tightly regulated Fe and Cu metabolism and interconnected antioxidant defense systems. These include redox-competent metal chaperons (e.g., the major Fe storage protein ferritin heavy chain (FTH1) and Cu-containing ceruloplasmin (CP) that both convert redox-active Fe^2+^ into redox-inactive Fe^3+^), selenoproteins (e.g., selenium-transporter selenoprotein P that sustains antioxidative selenoenzymes) and redox-buffering metalloenzymes (e.g., Cu/Zn/Mn superoxide dismutase (SOD) and selenium-containing glutathione peroxidase (GPX) that break down superoxide anions and lipid peroxides [16,17]). More specifically, alterations of Fe and Cu homeostasis are linked to the formation and propagation of AD-associated β-amyloid (Aβ) plaques [18,19], while redox-active metals complexed to Aβ-peptides can significantly enhance ROS generation [20,21], thus forming a vicious cycle that promotes disease progression.

Mn is also a redox-active element, but due to a higher reduction potential it is less prone to such redox chemistry compared to Fe and Cu. Importantly, Mn plays a key role in cellular adaption to OS, either through activating Mn-containing SOD enzymes and forming unbound free Mn-based antioxidants [22]. Moreover, Mn is a co-factor for enzymes (e.g., glutamine synthetase, pyruvate carboxylase) that are in involved in energy production, neuro-glial cell functions and neurotransmitter synthesis [23,24]. However, excess and long-term exposure of Mn can result in pathologies of the CNS, in particular manganism, with symptoms resembling PD [25,26]. In this context, supraphysiological Mn doses are capable of depleting Fe levels via disrupting both Fe-S cluster and heme enzyme biogenesis, thus evoking mitochondrial dysfunction and OS [27].

In sharp contrast to these redox-cycling trace elements, the highly abundant Zn metal has no direct redox capacity and always exists as bi-valent Zn^2+^ in biological systems. In total, 90% of the total brain Zn is protein-bound, associated with sulfur in cysteine residues allowing its rapid dissociation dependent on the cellular redox status. Protein-bound Zn essentially contributes to maintain redox balance by limiting OS by protecting thiol groups from oxidation and regulating glutathione metabolism [28,29]. Only 10% of Zn in the brain exists as unbound “labile” Zn pool and is located in presynaptic vesicles regulating synaptic neurotransmission of glutamatergic neurons and serving as endogenous neuromodulator of various receptors (e.g., AMPA, NMDA, GABA) [28,29,30]. Due to the redox-inert nature, Zn has a rather low toxicity. However, Zn deficiency has been shown to affect neuronal cell death and is suggested as a risk factor for aging and neurodegenerative disorders, including AD [31,32]. 

Several studies have already investigated total element concentrations in CSF [33,34], serum and blood [35] or even in paired CSF/serum samples [36,37] and calculated the Q_(CSF/Serum)_ ratio reflecting the metal homeostasis across NB. Employing speciation analysis techniques to characterize element-carrying proteins and low molecular mass (LMM)-element species provide novel insights into metal neurochemistry and transport from blood to the CSF compartment and vice versa [38,39,40,41,42,43]. Size exclusion chromatography (SEC) is a chromatographic method in which metallo-proteins and LMM compounds in solution are separated by their size. Peak matching with established protein standards identifies respective metallo-proteins, while mass calibration of the SEC columns allows for size attachment to not standard-matched molecules. Notably, SEC ensures that complex-bound metals remain at their carrying proteins or LMM-ligands, which is a mandatory prerequisite in element speciation. Combining SEC separation with dynamic reaction cell (DRC)–ICP–mass spectrometry as the element-specific chromatographic detector provides a unique method to quantify unknown metallo-proteins. This speciation technique has already been successfully applied to various biological matrices in animal experiments, analyzing porcine liver extracts [44], mouse and rabbit blood [45,46], serum and tissues of Wistar rats [47,48], as well as human material, including breast milk [49], thyroid tissue extracts [50], CSF [51,52,53] and paired CSF/serum samples [38,40,41,42,54].

In this study, we utilize a SEC-ICP-DRC-MS approach to gain a deeper knowledge of Fe, Zn, Cu, and Mn and their relative abundancies in size fractions from 24 paired human serum/CSF samples and in previously published brain samples of Mn exposed rats [55,56].

## 2. Results and Discussion

### 2.1. Total Quantification and SEC-Separation of Metal Species in Serum and CSF Samples

CSF is rigidly regulated by NB to ensure a stable environment for neuronal function. Furthermore, it provides a sink for the elimination of solutes from the brain interstitium [9]. To characterize the trace element composition under physiological conditions, we first performed ICP-MS analysis using 24 paired serum and CSF samples drawn from healthy control persons of our previous study [41] (age range 30–85 years; median 48.5 years; female:male ratio = 1:1) (see details in Material and Methods). In line with previous studies, we observed that concentrations of all four trace elements in the CSF were significantly lower than in corresponding serum samples, reflected in a ratio Q_(CSF/Serum)_ < 1, highlighting a selective and controlled permeation by NB to avoid excessive exposure to neuronal tissue [37,40] (Table 1, Figure 1A–D).

Consistent with previous investigations of brain metal neurochemistry [57,58,59,60], our data revealed the highest concentrations for Fe (19 ± 6.7 µg/L), followed by Zn (18 ± 6.9 µg/L), Cu (17 ± 2.1 µg/L) and Mn (0.5 ± 0.2 µg/L) in healthy CSF samples (Figure 1A–D).

Next, we investigated in our paired serum/CSF samples the speciation and distribution of all trace elements using SEC-ICP-DRC-MS for sufficient separation of metallo-proteins and different labile LMM metal containing compounds from each other. In contrast to Roos et al. [38], who analyzed CSF/serum pairs with off-line SEC-ICP-MS having a limited 6–70 kDa separation range, we applied our previously developed method [41,42,54,61,62,63], providing a separation range from ~0.2 to 1200 kDa and direct online hyphenation to ICP-DRC-MS. This set-up resulted in nine size fractions of metal species: >1200 kDa; 400–600 kDa; 150–300 kDa; 90–120 kDa; 40–80 kDa; 2–25 kDa; 0.7–2 kDa; 0.15–0.65 kDa; inorganic fraction. Before investigating our samples, we passed authentic candidate proteins (see details in Material and Methods) through the same columns using the same chromatography conditions. Figure 1E–H show the relative trace element composition of size-fractionated serum and CSF samples. Notably, not every fraction showed values above the limit of quantification (LOQ) for all elements. LOQ for metal species in SEC-ICP-DRC-MS were calculated as 10 σ-criterion. They were 20 ng/L for Mn species, 25 ng/L for Fe species, 65 ng/L for Cu species, and 60 ng/L for Zn species. 

Iron (Fe), the most enriched CSF metal, was drastically decreased across NB from 1220 ± 293 (serum) to 19 ± 6.7 µg/L (CSF) (Figure 1A and Table 1). Relative distribution in Fe species revealed a significant decrease of the 400–600 kDa in CSF compared to matched serum samples. Standard matching identified ferritin within this fraction. This 450 kDa multimeric iron-storage protein is built from 24 self-assembling heavy (FTH1, 21 kDa) and light chains (FTL, 19 kDa) and safely stores up to 4500 Fe^3+^ ions [64,65]. Parallel to the ferritin decrease, we observed a significant increase in the 0.7–2 kDa fraction in CSF. Since no specific Fe-standard compound co-eluted together with this particular size fraction, the exact nature of Fe species in this fraction remains to be elucidated (Figure 1E). The decrease of high molecular mass protein Fe fractions across the NB with a simultaneously higher proportion of small Fe species in CSF coincides with a healthy barrier function. A fully functional NB is retaining high molecular mass compounds more effectively than small compounds. This is expressed as clinical measure “albumin concentration Q_Alb_ (C_(CSF)_/C_(serum)_)” ratio, where the target value for patients with age range applicable to our samples is < 6.5 × 10^−3^ [66,67], and which indeed was found in our sample pairs (5.35 × 10^−3^). 

Zinc (Zn), the second most abundant element in the CSF, was also decreased across NB from 704 ± 159 (serum) to 18 ± 6.9 µg/L (CSF) (Figure 1B and Table 1). Relative distribution in Zn species revealed a significant decrease of the 400–600 kDa in CSF compared to serum samples. Although Zn is found in approximately 10% (up to 300 metalloenzymes) of the human proteome [68], we could not identify a Zn protein standard matching to this molecular mass. One possible explanation could be that higher molecular multimeric Zn proteins were eluted in this fraction. Furthermore, we observed an increase of Zn concentrations in the 40–80 kDa fraction. Using human serum albumin (HSA, 68 kDa) standard matching and mass calibration, we identified albumin as a candidate protein. Indeed, HSA carries divalent metal ions and is known also as a minor Zn carrier [69]. Furthermore, it could be hypothesized that SOD (32.5 kDa) might contribute in some degree to this fraction, since it might also contain traces from smaller metal species between 32 and 40 kDa when considering typical peak tailing instead of strictly calculating molecular ranges related to peak maxima (Figure 1F).

Copper (Cu) was significantly decreased across NB from 1380 ± 560 (serum) to 17 ± 2 µg/L (CSF) (Figure 1C and Table 1). Relative distribution in Cu species revealed a significant decrease of the 400–600 kDa in CSF (Figure 1G). For peak identification, we used CP standard solutions, which clearly eluted in the 400–600 kDa fraction. Although the CP-monomer has a molecular weight of 151 kDa, we hypothesized that CP forms higher molecular quadromeric (4 × 151 kDa) complexes (personal communication H. Zischka, TUM). Furthermore, we observed a significant increase in the 40–80 kDa Cu fraction matching with the HSA standard. Similar to Zn, HSA is also reported as a minor Cu carrier [70]. 

Lastly, Manganese (Mn) was markedly depleted across NB from 1350 ± 311 (serum) to 500 ± 200 ng/L (CSF) (Figure 1D and Table 1). Relative distribution of Mn species in CSF revealed a decrease of the >1200 kDa Mn fraction. However, molecules eluting in this fraction are above the analytical separation range of the SEC column and could not be annotated to specific Mn proteins. Moreover, we observed a decrease of the 400–600 kDa Mn fraction in CSF compared to serum samples. By standard matching, we identified α2-macroglobulin, consisting of four identical subunits of 180 kDa. The difference in nominal molecular weight (725 kDa) and the calculated MW range of the SEC fraction (400–600 kDa) can be explained by interactions between this Mn-bound α2-macroglobulin and the stationary phase [71]. Furthermore, we showed a decrease of Mn in the 40–80 kDa fraction. In line with our previous report, we identified transferrin (MW 78 kDa) as candidate protein in this fraction using standard matching and mass calibration [43]. Mn^3+^ is known to use free binding sites at transferrin, competing with Fe^3+^ [72,73,74]. Lastly, we observed an increase of the 0.15–0.65 kDa fraction in CSF compared to matched serum samples (Figure 1H). In line with our previous finding using the ESI-FT-ICR-MS approach [43], we identified Mn-citrate by standard matching. Although being still under debate, the significant increase of Mn-citrate in CSF suggests an active transport across NB, such as the monocarboxylic transporters [75,76].

### 2.2. Correlation of Trace Elements in Serum and CSF Samples

Next, we further determined the interdependency between all four elements in serum and CSF samples using Pearson’s correlation analysis. We demonstrated a positive correlation between “Fe vs. Mn” (r = 0.5177, *p* = 0.0096; Figure 2A) and a positive correlation of “Zn vs. Mn” (r = 0.5367, *p* = 0.0069; Figure 2B) only in CSF, but not in matched serum samples. 

Our observed positive Fe/Mn relationship in the CSF compartment is intriguing. Both trace elements share a resemblance in their size, physico-chemical properties, absorption and transport pathways, which would imply a competitive behavior. However, previous studies highlighted that Mn uptake is not mediated by shared Fe/Mn transporters (e.g., DMT1) [77]. Although still debatable, animal studies demonstrated that Mn exposure increased brain Fe level by reducing its CSF clearance, suggesting that Fe and Mn synergistically interact during transfer from plasma to brain compartments [78]. In contrast, our observation of “Fe vs. Zn” correlation in CSF samples is truly unexpected and previously not reported. Despite that, no significant correlation could be observed in other element pairs in both sample types (Figure 3), the herein observed positive correlations could be due to the small sample size, thus needed to be validated in a much larger patient cohort in future studies.

### 2.3. SEC-Separation of Metal Species in Brain Extracts from Mn-Exposed Rats

Despite the low statistical power, the positive correlations of “Fe vs. Mn” and “Zn vs. Mn” in human CSF samples inspired us to re-analyze brain extracts from our previously published rat model of Mn exposure [55,61]. In brief, RjHan:SD rats received Mn-enriched fodder (500 mg/kg body weight, n = 6) and the control group (n = 6) a standard diet with 23 mg/kg fodder (control group) for 53 days [55]. Prepared brain extracts (see Material and Methods) were subjected to SEC-DRC-ICP-MS analysis for Mn, Fe and Mn. Compared to control animals, we observed in Mn-exposed brains a significant accumulation in the 400–600 kDa fraction, which corresponded to α2-macroglobulin standard matching (Figure 4A). Interestingly, analyzing iron-containing proteins revealed a suppression of the 400–600 kDa fraction, which we confirmed to largely contain ferritin-bound Fe (Figure 4B). This finding is consistent with our previous study, where we confirmed a marked suppression of FTH1, along with a shift in the Fe^2+^/Fe^3+^ towards Fe^2+^ in Mn-exposed rat and mouse brains compared to control animals [55,56]. Numerous studies have pointed to a link between Mn and Fe metabolism (reviewed in [79]). Indeed, Mn can directly alter Fe homeostasis by increasing the binding between iron regulatory proteins (IRP) to the iron-responsive elements (IREs). This leads to a translational suppression of neuroprotective FTH1, thus less safely stored protein-bound Fe^3+^ and more free redox-active Fe^2+^ amount which generates neurotoxic ROS [56].

In contrast to Fe, we did not observe any significant changes in the distribution of Zn-containing protein fractions in Mn-exposed rat brains (Figure 4C). This could be explained by the fact that the turnover of Zn in the brain is much slower compared to peripheral tissue (e.g., liver) and less prone to alterations [31].

## 3. Conclusions and limitations of the Study

In this study, we analyzed 24 matched and paired CSF/serum samples for total concentrations and for speciation of four essential trance elements (Fe, Zn, Cu and Mn). We revealed that all metals were significantly down-concentrated in CSF compared to corresponding serum samples and demonstrated Q_(CSF/Serum)_ ratios similar to those reported elsewhere [37]. Using our SEC-ICP-DRC-MS approach, we further sufficiently separated metallo-proteins and different labile LMM metal containing compounds from each other. However, limitations of this technique include the unspecific mechanism of separation (driven by the molecular weight of metallo-compounds), the limited number of theoretical plates (SEC can resolve only species that differ about two-fold in molecular weight [71,80]) and that SEC-separation in plasma is remarkably influenced by the use of different mobile phases [46]. Our data shows that Fe-bound ferritin was decreased in CSF, whereas a low-molecular weight (LMW) Fe fraction was relatively increased that needs additional studies to determine iron candidate species. While we identified the presence of quadromeric formed CP-complexes in the decreased 400–600 kDa Cu fraction of CSF samples, we failed to identify a candidate protein in the decreased 400–600 kDa Zn fraction. Furthermore, we validated that the increased 40–80 kDa Cu/Zn fraction in CSF contains albumin-bound divalent metals. For Mn, the α-2-macroglobulin containing 400–600 kDa fraction was significantly lower concentrated, whereas the LMW citrate Mn fraction represented the main Mn fraction in the CSF. We concluded that all four transition metals are strictly controlled at NB, resulting in down-concentrated high-molecular weight (HMW) species and LWM fractions that are partly enriched in the CSF compared to the serum compartment. It is tempting to speculate whether there might be other, yet unidentified metal regulators. Future studies combining proteomic analysis with element speciation approaches could be a fruitful and promising strategy for identifying them. Finally, the isolated positive correlation of “Fe vs. Mn” and “Zn vs. Mn” in CSF samples could be due to the relatively low sample size and needs to be validated in a higher sample CSF/serum cohort. However, Mn-induced changes in ferritin-bound Fe fraction could support the Mn/Fe relationship, while Zn brain metabolism seems to be relatively unaffected by chronic Mn exposure.

## 4. Materials and Methods

### 4.1. Patient Samples

The study was performed in agreement with the ethical standards set in the Helsinki’s Declaration and its later amendments and was approved by the Ethics committee of the Technische Universität Munich (TUM) under ethics file number AZ 9/15 S. Study participants gave informed consent to the use of samples for scientific purpose. Serum and CSF sample pairs were drawn from 24 patients with unspecific neurological complaints, such as headache, dizziness and various sensory symptoms. CSF and serum samples were collected initially for our previous study [41] (see also Results and Discussion, first paragraph) and were handled according to clinical standard procedures [81]. In brief, CSF was collected from each individual by standardized lumbar puncture, and serum was obtained from blood drawn from the cubital vein directly after the spinal tap. CSF and serum samples—being taken from the same individual—are referred to as “paired samples”. The selected 24 patients had unremarkable CSF test results. Therefore, CSF and serum samples were considered to originate from neurologically healthy individuals. The aliquoted, frozen-stored samples (long-term storage at −80 °C) were thawed at 4 °C in the refrigerator, vortexed and ready for further analysis. Repetitive analysis of a sample showed no differences in size distribution pattern in analogy to our previous work [40].

### 4.2. Chemicals

Proteins, peptides, amino acids and elemental standards with different molecular masses were purchased from Merck-Sigma-Aldrich, Deisenhofen, Germany for mass calibration of the SEC column by retention time determination. These compounds were Blue dextran: 2000 kDa; iron-loaded ferritin: 900 kD; α-2-macroglobuline: 620 kD; α-globuline: 150 kD; ceruloplasmin 151 kDa; arginase: 107 kD; transferrin: 78 kD; albumin: 68.5 kDa; α-lactalbumin: 18 kDa; l-thyroxine: 777 Da; glutathione disulfide (ox.; GSSG): 612 Da; glutathione (red; GSH) 308 Da; Mn(II) (55 Da) = as inorganic MnCl_2_. TRIS, HNO_3,_ HCl (suprapure), NH_4_-acetate (NH_4_Ac) and acetic acid (HAc), MnCl_2_ * 5 H_2_O and citrate were ordered from Merck, Darmstadt, Germany. HNO_3_ was purified by subboiling distillation. Argon (Ar)_liqu_ and NH_3_ were purchased from Air-Liquide, Gröbenzell, Germany. An Ar vaporizer at the tank provided Ar gas. The *TSK*gel (230–450 mesh) for SEC separation was purchased from Merck, Darmstadt, Germany.

### 4.3. Standard and Sample Preparation

Stock solutions of MnCl_2_, FeCl_2,_ CuCl_2_ and ZnCl_2_ were prepared by dissolving 100 mg/L (related to respective metal). Mn-citrate stock solution was prepared by mixing a solution of 1 g/L citrate with a MnCl_2_ solution (5 mg/L) using a ratio of 4 + 1 (v + v), resulting in a Mn-citrate stock concentration of 1 mg Mn/L. Mn-albumin and Mn-transferrin stock solutions were prepared in analogy by mixing 1 g/L protein solution with 5 mg/L MnCl_2_ solution (4 + 1, each), resulting in 1 mg Mn/L for each compound. Stock solutions were aliquoted and stored in the dark at −20 °C. No destabilization of standard compounds was observed using these conditions. Working solutions were prepared daily by appropriate dilution with TRIS-HCl, 10 mM, pH 7.4. Single standards and the analysis of standard mixtures were used to achieve information on SEC retention times or migration times in CE separations. 

### 4.4. SEC Parameters

Size exclusion chromatography for species fractionation was chosen to avoid or minimize species changes during HPLC separation. SEC of Cu, Fe, Mn and Zn species from paired serum and CSF samples were performed by a hyphenated system consisting of a Knauer 1100 Smartline inert Series gradient HPLC system with an electronic sample injection valve with a 100 µL injection loop (Perkin Elmer, Rodgau-Jügesheim, Germany). Two serially installed SEC columns, a Biobasic 300 mesh column (300 × 8 mm ID, Thermo, separation range > 1200–5 kDa) was connected in front to a 550 × 10 mm ID Kronlab column filled with TSK-HW40S (separation range < 100–2000 Da). This column combination provided the necessary wide separation range, enabling the separation of various metalloprotein fractions from each other and from low-molecular weight metallo-compound fractions. Tris-HAc (10 mM, pH 7.4) + 250 mM NH_4_Ac was used as the eluent at a flow rate of 0.75 mL/min. In [41], the eluent already proved to be suitable with respect to species stability and minimized sticking of compounds to the stationary phase. For checking species stability, reinjection experiments were performed in analogy to [82]. The mass balance typically was between 91% and 108%. An auxiliary UV detector was installed between the outlet of the second column and the nebulizer of the ICP-MS. UV was detected at 220 and 280 nm. Thus, element (ICP-MS) and UV detection were possible in parallel with an insignificant time shift (UV to ICP-MS) of 15 s compared to void elution at 27 min (=1620 s) and full chromatograms of 70 min (=4200 s). Peak resolving in chromatograms for each investigated element was achieved by the selected column combination and for incomplete peak separation peak deconvolution using Peakfit^TM^ software was applied.

### 4.5. Columns Mass Calibration

The mass calibration in the serially connected columns was performed using proteins and LMM standards with known molecular weight. As mobile phase TRIS-HCl 10 mM, pH 7.4 + 250 mM NH_4_-Ac was used. The retention times (RT) were determined by peak maxima in UV and for metalloproteins additionally by ICP-MS detection. Retention times followed two calibration curves for the two columns: from 27 (void, >1.200 kDa) to 47 min (~5 kDa) according to the equation “ln(kDa) = −0.0716 * RT + 5.0939 (r^2^ = 0.9992) and from 48 (>2 kDa) to 60 min (^55^Mn^2+^, ^56^Fe^2+^) according to the equation “ln(kDa) = −0.2387 * RT +12.59 (r^2^ = 0.9993). Peak fractions from samples were assigned to molecular masses calculated due to the above equations.

### 4.6. ICP-MS Parameters

A NexIon ICP-MS, Perkin Elmer (Sciex, Toronto, Canada) with dynamic reaction cell capability was employed for online determination of ^55^Mn, ^57^Fe, ^63^Cu and ^66^Zn in the time-resolved mode. For SEC coupling the column effluent was passing the UV detector and then was directed to a Meinhard nebulizer (which was mounted to a cyclone spray chamber) using a PEEK transfer tube (ID 100 µm). The RF power was set to 1250 W, the plasma gas was 15 L Ar/min. The nebulizer gas was optimized and finally set to 0.98 (Meinhard) or 1.02 (Micromist) mL Ar/min. The dwell time was 500 ms. The dynamic reaction cell (DRC) was operated using NH_3_ as DRC gas, finally at a flow rate of 0.58 mL/min. The DRC bandpass (q) was set to 0.45. These parameters were the optimal conditions for this instrument. LOQ for metal species in SEC-ICP-DRC-MS were calculated as 10 σ-criterion. They were 20 ng/L for Mn species, 25 ng/L for Fe species, 65 ng/L for Cu species and 60 ng/L for Zn species.

### 4.7. ICP-OES

For quality control total element concentrations were additionally measured with an ICP-OES “Optima 7300” (Perkin Elmer) in parallel to ICP-DRC-MS. Sample introduction was performed by the instruments peristaltic pump at 1.0 mL/min and a Meinhard nebulizer which was mounted to a cyclon spray chamber. The measured spectral element lines were Cu: 324.754 nm; Fe: 259.941; Mn: 257.610 nm; Zn: 213.856 nm. The RF power was set to 1000 W, the plasma gas was 15 L Ar/min, whereas the nebulizer gas was 600 mL Ar/min. Regularly after 10 measurements three blank determinations and a control determination of certified element standards (SPEX) were performed. Comparability between the ICP-OES and -MS was checked by analyzing identical rat brain extracts (n = 10) and serum with both detectors in analogy to [83]. Results showed no significant differences between both metal detectors. The following recoveries (rec.) in ICP-DRC-MS detection were found related to ICP-OES (set 100% ) in rat serum and rat brain extracts: Cu Serum (~60 µg/L): 102 ± 20% rec.; Cu brain extract (~600 µg/kg): 103 ± 14% rec; Fe Serum (~2.5 mg/L): 128 ± 13% rec.; Fe brain extract (~6 mg/kg): 130 ± 14% rec; Mn Serum (~7 µg/L): 101 ± 7% rec.; Mn brain extract (~150 µg/kg): 102 ± 13% rec.; Zn Serum (~1450 µg/L): 82 ± 13% rec.; Zn brain extract (~5 mg/kg): 101 ± 14% rec.

### 4.8. Calculations and Statistical Analysis

The relative percentage changes of element species fractions were calculated in relation to the total element concentration changes using Microsoft Excel. The relationships between indicated trace elements in serum and CSF were determined using Pearson’s correlation coefficient, simple comparisons were analyzed using unpaired *t*-test and for multiple comparisons, two-way ANOVA followed by Bonferroni post hoc analysis was applied [84]. Statistical significance was calculated using GraphPad Prism software (Version 8.3.0). Significance was defined as *p* < 0.05 and values are reported as mean ± SEM. Further details of statistical tests are provided in the figure legends.

## Figures and Tables

**Figure 1 ijms-22-08892-f001:**
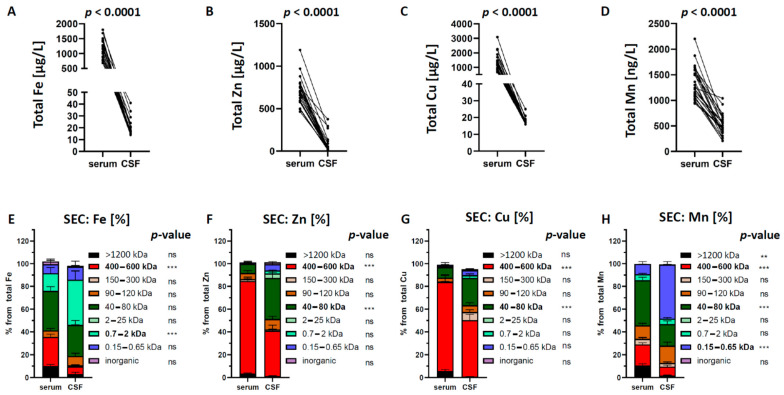
Changes of the total transition metal concentrations and of their relative distribution in size fractions from serum to corresponding CSF. (**A**–**D**) All four trace elements were significantly lower in CSF than in corresponding serum samples. (**E**) Changes of the Fe concentrations show a decrease for the 400–600 kDa and an increase for the 0.7–2 kDa fraction in CSF. (**F**) Changes of the Zn concentrations show a decrease for the 400–600 kDa and an increase for the 40–80 kDa fraction in CSF. (**G**) Changes of the Cu concentrations show a decrease for the 400–600 kDa and an increase for the 40–80 kDa fraction in CSF. (**H**) Changes of the Mn concentrations show a decrease for the >1200 kDa and the 400–600 kDa and an increase for the 0.15–0.65 kDa fraction in CSF. Differences were calculated using two-way ANOVA followed by Bonferroni post hoc analyses (** *p* < 0.01; *** *p* < 0.001). ns = not significant.

**Figure 2 ijms-22-08892-f002:**
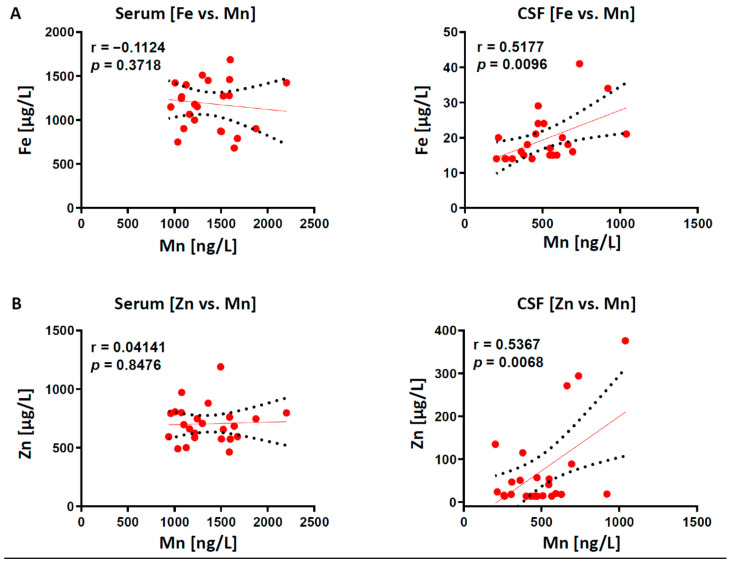
Pearson’s correlation analyses are shown for “Fe vs. Mn” and “Zn vs. Mn” interactions in 24 matched serum and CSF samples. (**A**) “Fe vs. Mn” is positively correlated only in CSF, but not in matched serum samples. (**B**) “Zn vs. Mn” is positively correlated only in CSF, but not in matched serum samples. “r” marks the linear Pearson correlation coefficient.

**Figure 3 ijms-22-08892-f003:**
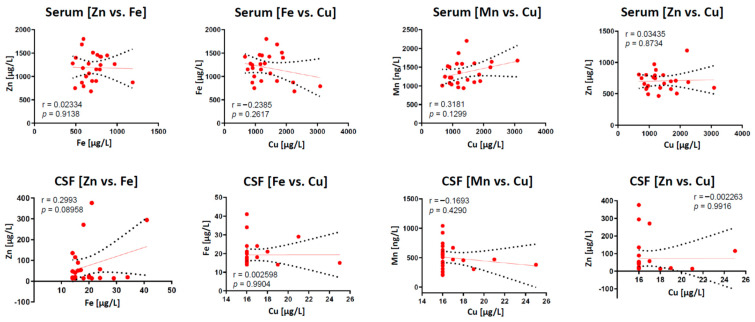
Pearson’s correlation analyses are shown for element vs. element interactions in serum and CSF samples. No correlation found for any shown element pairs, neither in serum nor CSF. r marks the Pearson correlation coefficient.

**Figure 4 ijms-22-08892-f004:**
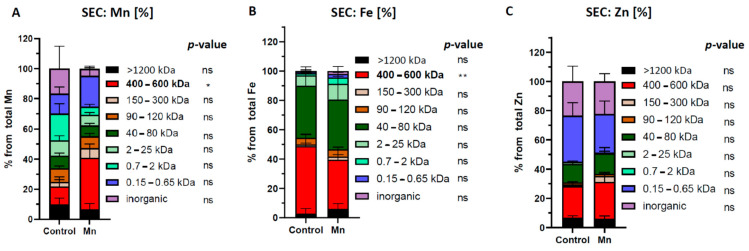
Comparison of relative element distribution in size fractions from brain extracts of control or orally Mn-exposed rats (backup samples from [55] were used). (**A**) Following Mn exposure only the Mn concentration of the 400–600 kDa fraction (containing α-2-macroglobuline) is significantly increased compared to controls (n = 6). (**B**) In Mn-treated brain samples the Fe concentration in the 400–600 kDa fraction (containing ferritin) is significantly decreased compared to controls (n = 6). (**C**) No significant changes between control and Mn-exposed brains were observed in Zn-containing fractions (n = 6). Differences were calculated using two-way ANOVA followed by Bonferroni post hoc analyses (* *p* < 0.05; ** *p* < 0.01). ns = not significant.

**Table 1 ijms-22-08892-t001:** Summary of total element concentrations in CSF and serum samples (mean ± SD), ratio of element concentrations. Q_(CSF/serum)_ and comparison with Q_(CSF/serum)_ values from [37]. Q_ctrl_ = ratio from control samples, Q_AD_ = ratio from Alzheimer’s disease samples.

	Mn (µg/L)	Fe (µg/L)	Cu (µg/L)	Zn (µg/L)
CSF (mean ± SD)	0.5 ± 0.2	19.4 ± 6.7	17.0 ± 2.0	18 ± 6.9
Serum (mean ± SD)	1.35 ± 0.3	1220 ± 293	1380 ± 560	704 ± 159
Q _(CSF/Serum)_ (this paper)	0.37	0.016	0.012	0.026
Q _ctrl (CSF/Serum)_ (ref. [37])	0.69 (0.26–1.87)	0.14 (0.06–0.38)	0.02 (0.01–0.03)	0.02 (0.01–0.20)
Q _AD (CSF/Serum)_ (ref. [37])	0.45 (0.08–1.50)	0.13 (0.04–0.52)	0.01 (0.01–0.11)	0.02 (0.00–0.19)

## Data Availability

Data is contained within the article or Appendix A. The entire data presented in this study are available in this article.

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
