# Peer review of "Simultaneous Quantification and Speciation of Trace Metals in Paired Serum and CSF Samples by Size Exclusion Chromatography–Inductively Coupled Plasma–Dynamic Reaction Cell–Mass Spectrometry (SEC-DRC-ICP-MS)"

_ijms, 2021, doi:10.3390/ijms22168892_

Round 1
Reviewer 1 Report
The transition metals, such as Fe, Zn or Cu are strictly controlled at neural barriers of neurologic patients. They exist in multiple oxidation states and participate in electron transfer reactions that are fundamental to sustain life for all organisms. The CNS is particularly sensitive to disturbances in the concentration of trace elements due to high energy demand of neurons. Given how relatively narrow is the range between deficiency and toxicity, trace elements must be tightly regulated to ensure healthy brain function is maintained. The regulation of their flow from the vascular system to the CNS is carried out by the BBB and BCB. At the same time, redox elements can catalyze reactive oxygen species, which can result in oxidative damage that is strongly related to e.g. with cell death or serious neurodegenerative diseases.
The analysis of the Mn, Cu, Fe and Zn content in physiological conditions was performed using the ICP-MS technique on samples of serum and cerebrospinal fluid taken from healthy controls. The results of these studies have already been published. In submitted manuscript they present the characteristics of selected trace elements on both sides of the neural barriers. Particular emphasis was placed on the relative change of the element fractions in relation to the total change in their concentration in neural barriers, which resulted in the percentage distribution of the fractions in the serum and CSF.
The study used the size exclusion chromatography (SEC) method, which allowed for the initial speciation of elements. The combination with dynamic reaction cell-ICP-MS as an element-specific chromatographic detector provided an appropriate method for analyzing the size fractions in quantification serum and CSF samples.
Taking into account the significance of the presented research, the presented manuscript can be accepted for publication after small additions, such as:
- There is no information about the LOD and LOQ values determined in the method validation process. “…not each fraction showed values above limits of quantification (LOQ) for all elements.” – what values?
- Usually, at the end of the manuscript, authors summarize the most important results, objectives and novelty of the work. There are no conclusions in this case.
Author Response
We thank all reviewers for their valuable comments.
We answer the specific reviewer questions and comments below point by point.
Following the recommendation of at least one of reviewers re-structuring and re-numbering of figures and introduction of a table is now included in the revision:
Table 1 is included now.
Previous Supplementary Figure 1 is now included as regular figure 3 as requested, previous Supplemetary Figure 2 is dropped.
As a consequence previous Figure 3 is now Figure 4, but extended with data on zinc, as requested by a reviewer.
Changes made due to recommendations are marked in blue.
According to reviewers´ recommendation we improved readability and polished for better understanding also for readers outside the field. The manuscript finally was checked for readability, understandability and language by a native speaker.
Specicic point to point response to reviewer:
Taking into account the significance of the presented research, the presented manuscript can be accepted for publication after small additions, such as:
- There is no information about the LOD and LOQ values determined in the method validation process. “…not each fraction showed values above limits of quantification (LOQ) for all elements.” – what values?
Thank you for this positive evaluation.
The requested information is/was already written at page 9 as follows:
LOQ for metal species in SEC-ICP-DRC-MS were calculated as 10 σ-criterion. They were: 20 ng/L for Mn-species, 25 ng/L for Fe-species, 65 ng/L for Cu-species and 60 ng/L for Zn-species.
Instrumental LOD are not reported as quantified values are relevant for the considerations in this paper and no values are reported which could not be quantified.
LOQ are now also provided at page 5, when the first time is referred to LOQ.
- Usually, at the end of the manuscript, authors summarize the most important results, objectives and novelty of the work. There are no conclusions in this case.
We have added a conclusion as suggested by the reviewer.

Reviewer 2 Report
This article is reporting analysis of trace metal (iron, zinc, copper and manganese) in serum and cerebrospinal fluid (CSF) by size exclusion chromatography coupled with mass spectrometry.
This straightforward analytical methodology reported by authors verified that total element concentration were significantly lower in CSF compared to serum. This correlative analysis will constitute the important goals and novelty of this paper.
The following suggested changes and recommendations should be introduced before the publication of the manuscript.
- Title: Specific suggestion to highlight importance of this topic is to insert the words “quantification and“ before speciation.
- Page 2, line 79, the references [10,11] should be moved to line 76 after “lipid peroxidases)”:
- Page 3, line 127, Fig 1 from page 5 should be moved here, immediately after (Figure 1A-D).
- Page 6, line 243. (Supplemental Figures S1) should be inserted here for the better visual correlation.
- Page 10, line 411. Please insert the literature reference for Bonferroni post hoc analysis.
- The last suggestion: For the better follow-up to analyze correlation of the concentration of trace metals in serum and CSF the authors should consider to insert the simple table with four columns with concentration of trace metals and two rows for serum and CSF. It will dramatically enhance the quick analysis and additionally highlights the significant differences without extensive search of the manuscript text.
The manuscript is of very good quality and urgent importance and is comprehensively well written and edited in order to meet the standard for the articles published in International Journal of Molecular Sciences. Thus, I certainly recommend it for publication after the correction of these suggested minor changes.
Author Response
We thank all reviewers for their valuable comments.
We answer the specific reviewer questions and comments below point by point.
Following the recommendation of at least one of reviewer re-structuring and re-numbering of figures and introduction of a table is now included in the revision:
Table 1 is included now.
Previous Supplementary Figure 1 is now included as regular figure 3 as requested, previous Supplemetary Figure 2 is dropped.
As a consequence previous Figure 3 is now Figure 4, but extended with data on zinc, as requested by a reviewer.
Changes made due to recommendations are marked in blue.
According to reviewers´ recommendation, we improved readability and polished for better understanding also for readers outside the field. The manuscript finally was checked for readability, understandability and language by a native speaker.
Specicic point to point response to reviewer:
The following suggested changes and recommendations should be introduced before the publication of the manuscript.
- Title: Specific suggestion to highlight importance of this topic is to insert the words “quantification and“ before speciation.
Thank you for your positive evaluation.
We have added “quantification and“ at requested title position.
- Page 2, line 79, the references [10,11] should be moved to line 76 after “lipid peroxidases)”:
We have moved the references as requested.
- Page 3, line 127, Fig 1 from page 5 should be moved here, immediately after (Figure 1A-D).
We have moved the figure and legend as requested.
- Page 6, line 243. (Supplemental Figures S1) should be inserted here for the better visual correlation.
We have inserted the supplemental figure1 as new figure 3 and renumbered the following figures.
- Page 10, line 411. Please insert the literature reference for Bonferroni post hoc analysis.
We added reference: Shi-Yi Chen, Zhe Feng, and Xiaolian Yi, A general introduction to adjustment for multiple comparisons, J Thorac Dis. 2017 Jun; 9(6): 1725–1729.
- The last suggestion: For the better follow-up to analyze correlation of the concentration of trace metals in serum and CSF the authors should consider to insert the simple table with four columns with concentration of trace metals and two rows for serum and CSF. It will dramatically enhance the quick analysis and additionally highlights the significant differences without extensive search of the manuscript text.
We have added the table 1 as requested.
The manuscript is of very good quality and urgent importance and is comprehensively well written and edited in order to meet the standard for the articles published in International Journal of Molecular Sciences. Thus, I certainly recommend it for publication after the correction of these suggested minor changes.
Thank you for your positive evaluation.
Reviewer 3 Report
The authors have carried out a large and important work, which allows, on the basis of a quantitative analysis of the content of transition metals entering the body of animals, to judge their distribution in the body. The results seem to be very interesting and will undoubtedly be in demand by the scientific community. Meanwhile, I have doubts that the authors need this particular journal to publish their results, and not a more specialized one. In addition, given that the authors are not the first and far from the only ones who conducted such studies, I think that in the introductory part it is necessary to give a more detailed comparative analysis of the studies available in the literature with a detailed discussion of all the advantages and disadvantages of previous studies. At the same time, I note that in this article there are no conclusions and a detailed discussion of the discovered patterns with their linkage with biochemical processes occurring in the body of laboratory animals after the introduction of metal salts. I believe that the article can be accepted for publication after substantial revision.
Author Response
We thank all reviewers for their valuable comments.
We answer the specific reviewer questions and comments below point by point.
Following the recommendation of at least one of reviewer re-structuring and re-numbering of figures and introduction of a table is now included in the revision:
Table 1 is included now.
Previous Supplementary Figure 1 is now included as regular figure 3 as requested, previous Supplemetary Figure 2 is dropped.
As a consequence, previous Figure 3 is now Figure 4, but extended with data on zinc, as requested by a reviewer.
Changes made due to recommendations are marked in blue.
According to reviewers´ recommendation, we improved readability and polished for better understanding also for readers outside the field. The manuscript finally was checked for readability, understandability and language by a native speaker.
Specicic point to point response to reviewer:
Meanwhile, I have doubts that the authors need this particular journal to publish their results, and not a more specialized one.
Considering this reviewer´s opinion the authors disagree, even the more, since this paper and its topic have been invited and the inviting editors knew the topic of this manuscript.
In addition, given that the authors are not the first and far from the only ones who conducted such studies, I think that in the introductory part it is necessary to give a more detailed comparative analysis of the studies available in the literature with a detailed discussion of all the advantages and disadvantages of previous studies.
We performed intense literature recherché about trace elements in CSF as well as SEC-ICP-DRC-MS applied to CSF or paired serum-cerebrospinal fluid samples. SEC-ICP-MS technique has been applied mainly to various other sample types and only very few papers (except ones from our group) are related to SEC-ICP-DRC-MS and application to paired CSF/serum samples. We have added respective references and shortly discussed the proven potential in such studies. We also discuss now more intensively the potential but also limitation of the SEC-ICP-DRC-MS technique.
A complete comparative literature evaluation however is not in line of this study since this is not a review article.
At the same time, I note that in this article there are no conclusions and a detailed discussion of the discovered patterns with their linkage with biochemical processes occurring in the body of laboratory animals after the introduction of metal salts. I believe that the article can be accepted for publication after substantial revision.
We have added a conclusion section and completed the discussion about their linkage with biochemical processes. Overall, the manuscript has undergone a very extensive revision.
Reviewer 4 Report
The paper by Michalke and coworkers deals with an interesting approach relying on the use of SEC-DRC-ICP-MS for the speciation of physiologically relevant metals in serum and CSF fluids.
This kind of studies are of interest and may also have a high potential in terms of translational medicine because can be directly applied to complex matrix ensuring a high sensitivity.
In this frame the present paper results of interest for a broad readership. Despite the above consideration the paper suffer of flaws, and for this I suggest major revision.
-typos are present in the whole ms, please revise accordingly
- "NB" should be defined in the abstract
-more important, the ms appears as a series of similar experiments for the detection of different metals / bound metals. I suggest authors to better discuss the implications of the reported findings. For instance, in section 2.3 authors deepen the positive correlation between Mn and Fe introducing animal models. However the same discussion / approach is not presented for the other positive correlation (Zn vs Mn).
-Also, the correlation between Fe and Zn could be of potential interest because of the role of Zn on the activity of neurotrophins. Pairwise the accumulation of Fe could be related to neurodegenerative diseases.
-The practical application of the method should be better discussed even adding a "conclusions section"
-In the present version the paper is not well readable for people not working in the field
Author Response
We thank all reviewers for their valuable comments.
We answer the specific reviewer questions and comments below point by point.
Following the recommendation of at least one of reviewer re-structuring and re-numbering of figures and introduction of a table is now included in the revision:
Table 1 is included now.
Previous Supplementary Figure 1 is now included as regular figure 3 as requested, previous Supplemetary Figure 2 is dropped.
As a consequence, previous Figure 3 is now Figure 4, but extended with data on zinc, as requested by a reviewer.
Changes made due to recommendations are marked in blue.
According to reviewers´ recommendation, we improved readability and polished for better understanding also for readers outside the field. The manuscript finally was checked for readability, understandability and language by a native speaker.
Specicic point to point response to reviewer:
The paper by Michalke and coworkers deals with an interesting approach relying on the use of SEC-DRC-ICP-MS for the speciation of physiologically relevant metals in serum and CSF fluids.
This kind of studies are of interest and may also have a high potential in terms of translational medicine because can be directly applied to complex matrix ensuring a high sensitivity.
In this frame the present paper results of interest for a broad readership. Despite the above consideration the paper suffer of flaws, and for this I suggest major revision.
-typos are present in the whole ms, please revise accordingly
We checked for typos and removed them.
- "NB" should be defined in the abstract
Is now defined.
-more important, the ms appears as a series of similar experiments for the detection of different metals / bound metals. I suggest authors to better discuss the implications of the reported findings. For instance, in section 2.3 authors deepen the positive correlation between Mn and Fe introducing animal models. However the same discussion / approach is not presented for the other positive correlation (Zn vs Mn).
We have added data now also for (Zn vs Mn) from the animal experiment and discuss them.
-Also, the correlation between Fe and Zn could be of potential interest because of the role of Zn on the activity of neurotrophins.
Fe and Zn did not show significant correlation
Pairwise the accumulation of Fe could be related to neurodegenerative diseases.
We have added some discussion points to that.
-The practical application of the method should be better discussed even adding a "conclusions section"
We have added a small para about application as well as a conclusion.
-In the present version the paper is not well readable for people not working in the field
We have restructured the text and better explained for better readability and understanding. A native speaker not working in the speciation field, has checked the ms for readability and understanding.
Round 2
Reviewer 3 Report
The manuscript has undergone a very extensive revision and I believe that in the present form manuscript can be published in IJMS.
Reviewer 4 Report
Authors properly solved concerns. The paper is suitable for publication.